# SHREWD:
# SEMANTIC HIERARCHY-BASED RELATIONAL EMBEDDINGS FOR WEAKLY-SUPERVISED DEEP HASHING

**Heikki Arponen and Tom E Bishop**
Intuition Machines Inc.
{heikki,tom}@intuitionmachines.com

## ABSTRACT

Using class labels to represent class similarity is a typical approach to training deep hashing systems for retrieval; samples from the same or different classes take binary 1 or 0 similarity values. This similarity does not model the full rich knowledge of semantic relations that may be present between data points. In this work we build upon the idea of using semantic hierarchies to form distance metrics between all available sample labels; for example cat to dog has a smaller distance than cat to guitar. We combine this type of semantic distance into a loss function to promote similar distances between the deep neural network embeddings. We also introduce an empirical Kullback-Leibler divergence loss term to promote binarization and uniformity of the embeddings. We test the resulting SHREWD method and demonstrate improvements in hierarchical retrieval scores using compact, binary hash codes instead of real valued ones, and show that in a weakly supervised hashing setting we are able to learn competitively without explicitly relying on class labels, but instead on similarities between labels.

## 1 INTRODUCTION

Content-Based Image Retrieval (CBIR) on very large datasets typically relies on hashing for efficient approximate nearest neighbor search; see e.g. Wang et al. (2016) for a review. Early methods such as (LSH) Gionis et al. (1999) were data-independent, but Data-dependent methods (either supervised or unsupervised) have shown better performance. Recently, Deep hashing methods using CNNs have had much success over traditional methods, see e.g. Hashnet (Cao et al., 2017), DADH (Li et al., 2018). Most supervised hashing techniques rely on a pairwise binary similarity matrix $S = \{s_{ij}\}$, whereby $s_{ij} = 1$ for images i and j taken from the same class, and 0 otherwise.

A richer set of affinity is possible using semantic relations, for example in the form of class hierarchies. Yan et al. (2017) consider the semantic hierarchy for non-deep hashing, minimizing inner product distance of hash codes from the distance in the semantic hierarchy. In the SHDH method (Wang et al., 2017), the pairwise similarity matrix is defined from such a hierarchy according to a weighted sum of weighted Hamming distances.

In Unsupervised Semantic Deep Hashing (USDH, Jin (2018)), semantic relations are obtained by looking at embeddings on a pre-trained VGG model on Imagenet. The goal of the semantic loss here is simply to minimize the distance between binarized hash codes and their pre-trained embeddings, i.e. neighbors in hashing space are neighbors in pre-trained feature space. This is somewhat similar to our notion of semantic similarity except for using a pre-trained embedding instead of a pre-labeled semantic hierarchy of relations.

Zhe et al. (2019) consider class-wise Deep hashing, in which a clustering-like operation is used to form a loss between samples both from the same class and different levels from the hierarchy.

Recently Barz & Denzler (2018) explored image retrieval using semantic hierarchies to design an embedding space, in a two step process. Firstly they directly find embedding vectors of the class labels on a unit hypersphere, using a linear algebra based approach, such that the distances of these embeddings are similar to the supplied hierarchical similarity. In the second stage, they train a standard CNN encoder model to regress images towards these embedding vectors. They do not consider hashing in their work.

## 2 FORMULATION

We also make use of hierarchical relational distances in a similar way to constrain our embeddings. However compared to our work, Barz & Denzler (2018) consider continuous representations and require the embedding dimension to equal the number of classes, whereas we learn compact quantized hash codes of arbitrary length, which are more practical for real world retrieval performance. Moreover, we do not directly find fixed target embeddings for the classes, but instead require that the neural network embeddings will be learned in conjunction with the network weights, to best match the similarities derived from the labels. And unlike Zhe et al. (2019), in our weakly supervised SHREWD method, we do not require explicit class membership, only relative semantic distances to be supplied.

Let $(x, y)$ denote a training example pair consisting of an image and some (possibly weakly) supervised target y, which can be a label, tags, captions etc. The embeddings are defined as $\hat{z} = f_\theta(x)$ for a deep neural network $f$ parameterized by weights $\theta$. Instead of learning to predict the target $y$, we assume that there exists an estimate of similarity between targets, $d(y, y')$. The task of the network is then to learn this similarity by attempting to match $\|\hat{z} - \hat{z}'\|$ with $d(y, y')$ under some predefined norm in the embedding space.

While in this work we use class hierarchies to implicitly inform our loss function via the similarity metric $d$, in general our formulation is weakly supervised in the sense that these labels themselves are not directly required as targets. We could equally well replace this target metric space with any other metric based on for instance web-mined noisy tag distances in a word embedding space such as GloVe or word2vec, as in Frome et al. (2013), or ranked image similarities according to recorded user preferences.

In addition to learning similarities between images, it is important to try to fully utilize the available hashing space in order to facilitate efficient retrieval by using the Hamming distance to rank most similar images to a given query image. Consider for example a perfect ImageNet classifier. We could trivially map all 1000 class predictions to a 10-bit hash code, which would yield a perfect mAP score. The retrieval performance of such a "mAP-miner" model would however be poor, because the model is unable to rank examples both within a given class and between different classes (Ding et al., 2018). We therefore introduce an empirical Kullback-Leibler (KL) divergence term between the embedding distribution and a (near-)binary target distribution, which we add as an additional loss term. The KL loss serves an additional purpose in driving the embeddings close to binary values in order to reduce the information loss due to binarizing the embeddings.

We next describe the loss function, $L(\theta)$, that we minimize in order to train our CNN model. We break down our approach into the following 3 parts:

$$L(\theta) = L_{sim} + \lambda_1 L_{KL} + \lambda_2 L_{cls} \qquad (1)$$

$L_{cls}$ represents a traditional categorical cross-entropy loss on top of a linear layer with softmax placed on the non-binarized latent codes. The meaning and use of each of the other two terms are described in more detail below. Similar to Barz & Denzler (2018) we consider variants with and without the $L_{cls}$, giving variants of the algorithm we term SHREWD (weakly supervised, no explicit class labels needed) and SHRED (fully supervised).

### 2.1 SEMANTIC SIMILARITY LOSS

In order to weakly supervise using a semantic similarity metric, we seek to find affinity between the normalized distances in the learned embedding space and normalized distances in the semantic space. Therefore we define

$$L_{sim} = \frac{1}{B^2} \sum_{b,b'=1}^{B} \left| \frac{1}{\tau_z} \|\hat{z}_b - \hat{z}_{b'}\|_M - \frac{1}{\tau_y} d(y_b, y_{b'}) \right| w_{bb'}, \qquad (2)$$

where $B$ is a minibatch size, $\|\ldots\|_M$ denotes Manhattan distance (because in the end we will measure similarity in the binary space by Hamming distance), $d(y_b, y_{b'})$ is the given ground truth similarity and $w_{bb'}$ is an additional weight, which is used to give more weight to similar example pairs (e.g. cat-dog) than distant ones (e.g. cat-moon). $\tau_z$ and $\tau_y$ are normalizing scale factors estimated

| Method | mAP | mAHP@250 | Classification accuracy |
|---|---|---|---|
| DeViSE (Frome et al., 2013)[†] | 0.5016 | 0.7348 | 74.66% |
| Sun et al. (2017)[†] | 0.6202 | 0.7950 | **76.96%** |
| Barz & Denzler (2018)[†], $L_{CORR}$ | 0.5900 | 0.8290 | 75.03% |
| Barz & Denzler (2018)[†], $L_{CORR+CLS}$ | 0.6107 | 0.8329 | 76.60% |
| Zhe et al. (2019)[‡] | 0.8259[‡] | 0.8667[‡] | n/a |
| $L_{sim}$ only | 0.2204 | 0.7007 | 10.01% |
| $L_{cls}$ only | 0.5647 | 0.8124 | 73.00% |
| $L_{sim} + L_{cls}$ only | 0.5292 | 0.8188 | 69.68% |
| $L_{KL} + L_{cls}$ only | 0.3010 | 0.6215 | 69.25% |
| SHREWD [Ours] $L_{KL} + L_{sim}$ | 0.6514 | 0.8690 | 70.79% |
| SHRED [Ours] $L_{KL} + L_{sim} + L_{cls}$ | 0.7049 | **0.8802** | 72.77% |

Table 1: Retrieval Performance and Ablation Study on CIFAR-100, 64 bit hash codes. † indicates non-quantized embedding codes. ‡ mAHP@2500 measured with this method, so not equivalent. Note that while $L_{cls}$ performs best on supervised classification, $L_{sim}$ allows for better retrieval performance, however this is degraded unless $L_{KL}$ is also included to regularize towards binary embeddings. For measuring classification accuracy on methods that don't include $L_{cls}$, we measure using a linear classifier with the same structure as in $L_{cls}$ trained on the output of the first network.

from the current batch. We use a slowly decaying form for the weight, $w_{bb'} = \gamma^\rho / (\gamma + d(y_b, y_{b'}))^\rho$, with parameter values $\gamma = 0.1$ and $\rho = 2$.

## 2.2 KL-DIVERGENCE BASED DISTRIBUTION MATCHING LOSS

Our empirical loss for minimizing the KL divergence $KL(p||q) \doteq \int dz\, p(z) \log(p(z)/q(z))$ between the sample embedding distribution $p(z)$ and a target distribution $q(z)$ is based on the Kozachenko-Leonenko estimator of entropy Kozachenko & Leonenko (1987), and can be defined as

$$L_{KL} = \frac{1}{B} \sum_{b=1}^{B} \left[ \log\left(\nu(\hat{z}_b; z)\right) - \log\left(\nu(\hat{z}_b; \hat{z})\right) \right], \tag{3}$$

where $\nu(\hat{z}_b; z)$ denotes the distance of $\hat{z}_b$ to a nearest vector $z_{b'}$, where $z$ is a sample (of e.g. size $B$) of vectors from a target distribution. We employ the beta distribution with parameters $\alpha = \beta = 0.1$ as this target distribution, which is thus moderately concentrated to binary values in the embedding space. The result is that our embedding vectors will be regularized towards uniform binary values, whilst still enabling continuous backpropagation though the network and giving some flexibility in allowing the distance matching loss to perform its job. When quantized, the resulting embeddings are likely to be similar to their continuous values, meaning that the binary codes will have distances more similar to their corresponding semantic distances.

## 3 EXPERIMENTAL RESULTS

**Metrics** As discussed in section 2, the mAP score can be a misleading metric for retrieval performance when using class information only. Similarly to other works such as Deng et al. (2011); Barz & Denzler (2018), we focus on measuring the retrieval performance taking semantic hierarchical relations into account by the mean Average Hierarchical Precision (mAHP). However more in line with other hashing works, we use the hamming distance of the binary codes for ranking the retrieved results.

**CIFAR-100** We first test on CIFAR-100 Krizhevsky & Hinton (2009) using the same semantic hierarchy and Resnet-110w architecture as in Barz & Denzler (2018), where only the top fully connected layer is replaced to return embeddings at the size of the desired hash length. See Tables 1,2 for comparisons with previous methods, an ablation study, and effects of hash code length.

| Code length | mAP result | mAHP result | Classification accuracy |
|---|---|---|---|
| 16 bits | 0.3577 | 0.7478 | 65.65% |
| 32 bits | 0.5114 | 0.8202 | 65.00% |
| 64 bits | 0.6514 | 0.8690 | 70.79% |
| 128 bits | **0.6857** | **0.8760** | 70.29% |

Table 2: Effect of hash code length on CIFAR-100 for SHREWD

| Method | mAP | mAHP@250 | Classification accuracy |
|---|---|---|---|
| Barz & Denzler (2018) (floating point embeddings) | 0.3037 | 0.7902 | 48.97% |
| SHREWD [Ours] (64 bit binary) | 0.4604 | 0.8676 | — |
| SHREWD [Ours] (128 bit binary hash codes) | 0.5067 | 0.8674 | — |
| SHREWD [Ours] (floating point embeddings) | — | 0.8733 | 63.28% |
| SHRED [Ours] (64 bit binary) | **0.5594** | **0.8885** | — |

Table 3: Performance on ILSVRC 2012, floating point vs quantized hash codes (NB classifier is only trained by using floating point embeddings as features)

**ILSVRC 2012** We also evaluate on the ImageNet Large Scale Visual Recognition Challenge (ILSVRC) 2012 dataset. For similarity labels, we use the same tree-structured WordNet hierarchy as in Barz & Denzler (2018). We use a standard Resnet-50 architecture with a fully connected hashing layer as before. Retrieval results are summarized in Table 3. We compare the resulting Hierarchical Precision scores with and without $L_{KL}$, for binarized and continuous values in Figure 1. We see that our results improve on the previously reported hierarchical retrieval results whilst using quantized embeddings, enabling efficient retrieval.

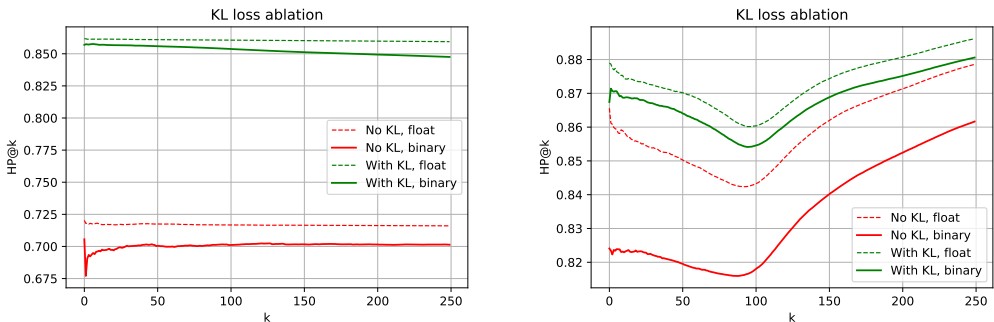

Figure 1: Hierarchical precision @k for CIFAR-100 (left) and ILSVRC-2012 (right) for 64-bit SHREWD. We see a substantial drop in the precision after binarization when not using the KL loss. Also binarization does not cause as severe a drop in precision when using the KL loss.

## 4 CONCLUSIONS

We approached Deep Hashing for retrieval, introducing novel combined loss functions that balance code binarization with equivalent distance matching from hierarchical semantic relations. We have demonstrated new state of the art results for semantic hierarchy based image retrieval (mAHP scores) on CIFAR and ImageNet with both our fully supervised (SHRED) and weakly-supervised (SHREWD) methods.

## 5 ACKNOWLEDGEMENTS

The authors would like to thank Eli-Shaoul Khedouri, Marcel Ackermann and Debajyoti Datta for useful discussions on these research subjects.

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
