# OpenReview forum: "SHREWD: Semantic Hierarchy Based Relational Embeddings For Weakly-Supervised Deep Hashing"
_ICLR.cc/2019/Workshop/LLD — LLD 2019_

### Official Review · AnonReviewer2 · 2019-04-07
**Interesting and excellent work**

**Rating:** 5
**Confidence:** 2

**Review:**

Summary: This paper introduces three loss terms (L_cls, L_KL, L_sim), and shows performances each loss term. This work achieves reasonable performances without explicit class labels. I accept this paper.

Notes:
- The paper introduces the semantic similarity loss which is to learn similarity in the embedding space given similarity between targets. In the ablation study, the loss function plays an important role in improving results.
- Also, they introduce the KL divergence term between the embedding distribution and binary target distribution. The paper also shows that the performances without the KL loss. With the KL loss, the proposed model shows substantial improvement.
- The paper did experiments on different hash code lengths and showed better performances on the longer hash code

I strongly accept this paper. This paper introduces novel loss functions and demonstrates that their method shows improvement in fully supervised and weakly supervised settings.

---

### Official Review · AnonReviewer1 · 2019-04-14
**Nice workshop contribution**

**Rating:** 4
**Confidence:** 2

**Review:**

The paper proposes to use semantic hierarchies to estimate distance metrics between sample labels instead of the standard 0/1 similarity values. The semantic distance is used as an additional supervision signal to regularize deep nets.

The topic is relevant and the paper could of interest to the workshop attendees. The approach is intuitive, described well, and the experimental results seem to be comprehensive (for the CIFAR-100 and ImageNet datasets they were carried on). It's also nice to see authors including an ablation study for their method.

---

### Decision · Program_Chairs · 2019-04-16
**Acceptance Decision**

Accept